# Virus Mimetic Poly (I:C)-Primed Airway Exosome-like Particles Enter Brain and Induce Inflammatory Cytokines and Mitochondrial Reactive Oxygen Species in Microglia

**DOI:** 10.3390/biology10121359

**Published:** 2021-12-20

**Authors:** Deimantė Kulakauskienė, Deimantė Narauskaitė, Dovydas Gečys, Otilija Juknaitė, Lina Jankauskaitė, Aistė Masaitytė, Jurgita Šventoraitienė, Hermanas Inokaitis, Zoja Miknienė, Ilona Sadauskienė, Giedrius Steponaitis, Zbigniev Balion, Ramunė Morkūnienė, Neringa Paužienė, Dainius Haroldas Pauža, Aistė Jekabsone

**Affiliations:** 1Institute of Pharmaceutical Technologies, Faculty of Pharmacy, Lithuanian University of Health Sciences, LT-50162 Kaunas, Lithuania; deimante.kulakauskiene@lsmu.lt (D.K.); deimante.narauskaite@lsmu.lt (D.N.); dovydas.gecys@lsmuni.lt (D.G.); otilija.juknaite97@gmail.com (O.J.); lina.jankauskaite@lsmuni.lt (L.J.); zbigniev.balion@lsmuni.lt (Z.B.); 2Department of Drug Chemistry, Faculty of Pharmacy, Lithuanian University of Health Sciences, LT-50162 Kaunas, Lithuania; ramune.morkuniene@lsmuni.lt; 3Laboratory of Molecular Cardiology, Institute of Cardiology, Lithuanian University of Health Sciences, LT-50162 Kaunas, Lithuania; 4Institute of Physiology and Pharmacology, Faculty of Medicine, Lithuanian University of Health Sciences, LT-50162 Kaunas, Lithuania; 5Department of Pediatrics, Faculty of Medicine, Lithuanian University of Health Sciences, LT-50161 Kaunas, Lithuania; 6Institute of Anatomy, Faculty of Medicine, Lithuanian University of Health Sciences, LT-44307 Kaunas, Lithuania; a.masaityte@gmail.com (A.M.); jurgita.sventoraitiene@lsmuni.lt (J.Š.); hermanas.inokaitis@lsmuni.lt (H.I.); neringa.pauziene@lsmuni.lt (N.P.); dainius.pauza@lsmuni.lt (D.H.P.); 7Large Animal Clinic, Faculty of Veterinary Medicine, Lithuanian University of Health Sciences, LT-47181 Kaunas, Lithuania; zoja.mikniene@lsmuni.lt; 8Laboratory of Molecular Neurobiology, Neuroscience Institute, Lithuanian University of Health Sciences, LT-50162 Kaunas, Lithuania; ilona.sadauskiene@lsmuni.lt; 9Laboratory of Molecular Neurooncology, Neuroscience Institute, Lithuanian University of Health Sciences, LT-50162 Kaunas, Lithuania; giedrius.steponaitis@lsmuni.lt; 10Laboratory of Biochemistry, Neuroscience Institute, Lithuanian University of Health Sciences, LT-50162 Kaunas, Lithuania; 11Laboratory of Preclinical Drug Investigation, Institute of Cardiology, Lithuanian University of Health Sciences, LT-50162 Kaunas, Lithuania

**Keywords:** airway cell exosomes, viral infection, microglia, mitochondria, reactive oxygen species

## Abstract

**Simple Summary:**

Upper respiratory tract viral infections are among the most common diseases. The blood-brain barrier protects the brain from direct invasion of pathogens. However, the cells share their content with other cells in small nanovesicles called exosomes that can travel long distances and cross biological barriers. Therefore, virus-infected cell extracellular vesicles (EVs) might transmit inflammatory signals or even viral particles to other cells. If they would carry such signals or particles to the central nervous system, it might cause neuroinflammation. However, the migration and impact of virus-primed airway cell EVs on the brain have not been studied yet. Therefore, the study aimed to track airway EVs from the respiratory tract to the brain and determine how infection-primed particles affect microglia—the cells responsible for immune response in the brain. The study revealed that airway cell EVs enter the brain within an hour and gather in microglia. Interestingly, many airway EVs were found in the hippocampus, the region most affected by Alzheimer’s disease. Moreover, EVs from virus-infected airway cells stimulated reactive oxygen species in microglia and induced other inflammation mediators in the brain. Thus, airway cells indeed might communicate inflammatory information to the brain during viral infection.

**Abstract:**

Viral infections induce extracellular vesicles (EVs) containing viral material and inflammatory factors. Exosomes can easily cross the blood-brain barrier during respiratory tract infection and transmit the inflammatory signal to the brain; however, such a hypothesis has no experimental evidence. The study investigated whether exosome-like vesicles (ELVs) from virus mimetic poly (I:C)-primed airway cells enter the brain and interact with brain immune cells microglia. Airway cells were isolated from Wistar rats and BALB/c mice; microglial cell cultures—from Wistar rats. ELVs from poly (I:C)-stimulated airway cell culture medium were isolated by precipitation, visualised by transmission electron microscopy, and evaluated by nanoparticle analyser; exosomal markers CD81 and CD9 were determined by ELISA. For in vitro and in vivo tracking, particles were loaded with Alexa Fluor 555-labelled RNA. Intracellular reactive oxygen species (ROS) were evaluated by DCFDA fluorescence and mitochondrial superoxide—by MitoSOX. ELVs from poly (I:C)-primed airway cells entered the brain within an hour after intranasal introduction, were internalised by microglia and induced intracellular and intramitochondrial ROS production. There was no ROS increase in microglial cells was after treatment with ELVs from airway cells untreated with poly (I:C). In addition, poly (I:C)-primed airway cells induced inflammatory cytokine expression in the brain. The data indicate that ELVs secreted by virus-primed airway cells might enter the brain, cause the activation of microglial cells and neuroinflammation.

## 1. Introduction

The upper respiratory tract is considered a gateway for viruses to enter the human body. Approximately 90% of the upper airway tract infections are caused by viruses [1], and over 200 different viruses have been isolated from the human upper respiratory tract [2]. Generally, respiratory viruses, such as rhinoviruses, influenza, adenoviruses, enteroviruses, does not spread to the distant tissues; however, accumulating evidence shows that viral material carried by extracellular vesicles (EVs) originating from infected cells might induce cellular alterations outside of the respiratory tract [3,4,5]. The blood-brain barrier protects the brain from direct invasion of pathogens. However, a class of EVs called exosomes can effectively cross the blood-brain barrier by adsorptive transcytosis, receptor-mediated and other suggested pathways and carry endogenous material to the brain from the peripheral circulatory system [6,7,8]. Exosomes are 30–150 nm diameter EVs formed by endosomal maturation pathways and essential for intercellular communication [8,9]. They are secreted by almost every cell type and present in various body fluids, including urine, blood, breast milk, and saliva [10]. Exosomes contain a variety of bioactive molecules such as proteins, lipids and several RNA species [11]. Studies have shown that exosomal cargo depends on the physiological state of the parental cell [3,12,13]. It was observed that exosome content alternates in response to viral infection and that exosomes isolated from infected cells can activate innate immune response [3]. Several studies have shown that viral material transported by EVs can induce immune response [14], viral replication [15], cell apoptosis, cytokine release suppression [16,17], as well as contribute to NLRP3 inflammasome and IL-1β production [18].

One of the most plausible exosome recipients in the brain are microglial cells. Microglia and non-parenchymal macrophages in the brain are mononuclear phagocytes that are essential in brain homeostasis. Microglia constitute 5–10% of total brain cells and are the only true CNS parenchymal macrophages interacting with neurons, astrocytes and oligodendrocytes [19,20]. Microglia are involved in synaptic plasticity by synaptic pruning in the developing and adult brain; they also phagocytose dying, dead, and sometimes healthy cells [21,22,23]. Additionally, microglia respond to an infection or brain damage by transforming into an active inflammatory phenotype and protecting the brain from pathogens. However, prolonged microglial activation might lead to neurodegenerative disease development [24]. Recent evidence indicates that peripheral infections can stimulate an immune response in the brain, causing irreversible genetic and epigenetic changes in brain immune cells leading to the formation of immune memory [25]. Antiviral immune response and immune memory formation are related to rearrangements in mitochondrial network and dynamics and involve reactive oxygen species (ROS) signalling [26]. Exosomes might carry viral particles and/or inflammatory molecules. One might speculate that exosomes produced during frequent recurrent viral respiratory infections might cause microglia activation and maintenance of immune reactivity in the brain. However, such a hypothesis has no experimental evidence yet. This study investigates whether exosomes from cultured airway epithelial cells and fibroblasts under simulated infection—treatment with virus mimetic Toll-like receptor-3 agonist poly (I:C)—can enter the brain after intranasal delivery and how they interact with brain immune cells microglia.

## 2. Materials and Methods

### 2.1. Experimental Design

The experimental design of the study is presented in Figure 1. First of all, airway cells were isolated from laboratory rodent lungs, cultivated and primed with poly (I:C). Then, ELVs were isolated, characterised and prepared for in vivo intranasal and in vitro cell culture introduction. In parallel, primary microglial culture was prepared. Next, the poly (I:C)-treated and untreated airway cell ELVs were introduced to cultivated microglia, and the cultures were monitored for particle internalisation and intracellular plus intramitochondrial ROS production. In addition, stained airway cell ELVs were intranasally introduced to laboratory mice, and particle localisation in immunostained microglial cells in the brain cryosections was monitored after 1, 2, 3, and 5 h. Finally, poly (I:C)-primed airway ELVs-treated brains and microglial cultures were evaluated for inflammatory cytokine expression by qRT-PCR.

### 2.2. Primary Culture of Airway Cells

Primary airway cells were prepared from 6–7-day-old Wistar rats for in vitro studies, and primary mouse airway cells for in vivo studies were made from 9–11 weeks old Balb/c mice. All experimental procedures were performed according to the Republic of Lithuania Law on the Care, Keeping and Use of Animals. The rodents were maintained and handled at the Lithuanian University of Health Sciences animal house in agreement with the ARRIVE guidelines.

For isolation of airway cells, the animals were sacrificed by cervical dislocation. The trachea and lungs were exposed and separated from the thorax, followed by removal of the heart, trachea and large bronchi. The remaining lung tissue was rinsed twice with PBS, transferred to DMEM, minced with sharp scissors and digested in 1% trypsin in DMEM solution for up to 30 min at 37 °C with gentle agitation. The digestion process was inactivated with the DMEM and 10% FBS. The preparation was filtered through the 70-µm cell strainer and centrifuged at 400× *g* for 10 min at 4 °C. The cells in the pellet were suspended in a growth medium (DMEM with GlutaMAX^TM^, 25 mM HEPES, 10% FBS and 1%, or 10,000 IU/mL, penicillin-streptomycin solution) and seeded in a T-75 flask. After 24 h, the medium was removed, centrifuged again and the cells in the pellet were grown in a T-75 flask for 6–9 d until used for further experiments. According to morphological evaluation under a brightfield microscope, the cultures comprised of fibroblasts and epithelial cells at an approximate ratio of 1:1 (Figure A1).

### 2.3. Cultures of Primary Mixed Glia and Pure Microglia

Mixed glial and pure microglial cultures were prepared from 6–7-day-old Wistar rat pups, as described in [27], with minor modifications. All animal care and procedures were performed according to the Republic of Lithuania Law on the Care, Keeping and Use of Animals following ARRIVE guidelines. Briefly, after decapitation, cerebral cortices were separated from the remaining parts of the brain and placed in Petri dishes containing PBS, glucose (13 mM), and penicillin-streptomycin solution (10,000 IU/mL–10,000 μg/mL). After removing the meninges, the cortical tissues were minced and transferred to the centrifugal tube with preheated (37 °C) Versene (Gibco™, Thermo Fisher Scientific, Bleiswijk, The Netherlands, 1:50,000) solution and incubated for 10 min in 37 °C with gentle agitation. After incubation, the solution was triturated with Pasteur pipettes and centrifuged at 290× *g* at room temperature. The pellet was resuspended in DMEM with GlutaMAX^TM^, (Thermo Fisher Scientific, Bleiswijk, The Netherlands) and 10% FBS, passed through 40-μm mesh, transferred to poly-L-lysine coated T-75 flasks and cultivated for 13 d replacing growth medium every 5th d. Microglial cells were detached by gently shaking the flask for 5 min on an orbital shaker, isolated by centrifugation at 290× *g* for 5 min and grown in T-75 flasks in the same growth medium.

### 2.4. Isolation of ELVs

Rat and mouse airway cells were cultured in T-75 flasks until reaching 70–80% confluency. The growth medium was replaced with DMEM without FBS to avoid contamination with EVs that are present in the serum. The cells were treated with 1 μg/mL poly (I:C) for 1 h, washed with a serum-free medium to remove any exosomes and poly I:C, and then further incubated for 24 h in a serum-free medium without poly (I:C). After collection, the cell-conditioned medium was passed through 0.22 µm PVDF filters. ELVs were isolated using Total Exosome Isolation Reagent (Invitrogen) according to the manufacturer’s manual. Briefly, conditioned media were mixed with the reagent at a ratio of 2:1. The mixture was incubated at 4 °C for 16 h and centrifuged at 10,000× *g* for 1 h at 4 °C. The pellets were resuspended in 250 µL of PBS, aliquoted and stored at −80 °C for further use.

### 2.5. Characterisation of ELVs

The amount of total protein in ELV samples was determined by Bradford assay (Sigma-Aldrich, Taufkirchen, Germany), measuring the absorption of λ = 595 nm light in a Tecan Infinite 200 PRO plate reader. The exosomal markers tetraspanins CD63 and CD9 were quantified by ELISA kits (Abbexa CD63 ELISA Kit and Abbexa MRP1 ELISA Kit, Cambridge, UK) according to the manufacturer’s instructions.

The nanoparticle size distribution in isolated particle preparations was determined by dynamic light scattering (ZetaSizer Nano ZS, Malvern PANalytical, Malvern, UK). In short, 50 µL of particle preparation was homogenised for 5 min using 30G needle. Ten microliters of preparation were mixed with 1990 μL of PBS in analytical cuvettes, monitored in the analyser, and the data were processed by ZetaSizer Nano software (Malvern PANalytical, Malvern, UK).

ELV imaging was performed by transmission electron microscopy. Isolated particles were homogenised for 5 min with 30G needle and mixed with 4% paraformaldehyde at the ratio of 1:1. This solution was applied on carbon-coated Formvar copper meshes FCFT200-Cu-50 200 MESH (Sigma-Aldrich, Taufkirchen Germany). The meshes were fixed in 1.7% glutaraldehyde solution for 5 min, washed twice in deionised water for 2 min, and stained with 2% uranyl acetate for 2 min. After staining, the meshes were incubated with freshly prepared 2.25% methylcellulose and 2% uranyl acetate in a *v*/*v* ratio of 4:1 for 10 min on an ice table. Prepared meshes were carefully dried on filter paper for 10 to 15 min and visualised using a transmission electron microscope Tecnai BioTwin Spirit G2 (FEI, Eindhoven, The Netherlands) on 80 kV voltage. Electron microscope images were taken with a bottom-mounted 16 MP TEM CCD camera Eagle 4K employing TIA (FEI, Eindhoven, The Netherlands).

### 2.6. ELV Labelling for In Vitro and In Vivo Tracking

ELVs were labelled with Alexa Fluor 555 dye (AF555) conjugated to oligonucleotides (BLOCK-it Alexa Fluor Red Fluorescent Control, Invitrogen, Thermo Fisher Scientific, Vilnius, Lithuania) by lipofection (RNAiMAX, Invitrogen, Thermo Fisher Scientific, Lithuania, Vilnius). Briefly, a mixture of 0.2 µM of AF555-oligonucleotide conjugate was mixed with 3 µL of RNAiMAX reagent in 100 µL of the Opti-MEM^TM^ medium (Gibco™, Thermo Fisher Scientific, Bleiswijk, The Netherlands) and incubated for 5 min at room temperature. Particle preparation (1 mg/mL of total protein) was added into lipofection mixture and incubated at 37 °C for one hour. After incubation, unincorporated dye and residual micelles were removed using Exosome Spin Columns (Invitrogen, Thermo Fisher Scientific, Vilnius, Lithuania). For a micelle-cleaning efficiency assessment, the fluorescence intensity of lipofection mix comprising 0.2 µM pmol AF555-oligonucleotide conjugate, 7.5 uL RNAiMAX reagent and 92.5 uL of PBS was measured before and after cleaning procedure using Tecan Infinite Pro plate reader. The calculated efficiency of unincorporated dye elimination from ELV samples was 99.99% (Figure A2). After the labelling and cleaning procedure, the particles were concentrated using 100 K Amicon^®^ ultra centrifugal filters (Merck Millipore, Darmstadt, Germany) and used for internalisation and tracking analysis.

### 2.7. Intranasal In Vivo Administration of Airway ELVs

All experimental procedures were performed on 4-month-old Balb/c mice according to the Law of the Republic of Lithuanian Animal Welfare and Protection (License of the State Food and Veterinary Service for working with laboratory animals No. G2-96). The mice were maintained and handled at the Lithuanian University of Health Sciences animal house in agreement with the ARRIVE guidelines. Before the intranasal administration of fluorescently labelled ELVs from poly (I:C)-primed and not primed airway cells, each nostril was treated with 100U hyaluronidase dissolved in 5 µL PBS to increase the permeability of the mucus. After 30 min, 25 μL of ELV solution containing 30 μg of total protein was introduced to each nostril (60 μg per mouse). The solution was administered gradually in 5 μL portions followed by a 5 min interval and alternating the nostrils. After 1, 2, 3, and 5 h, the mice were anaesthetised and sacrificed, and the brains were further processed for immunohistochemical analysis.

### 2.8. In Vitro Particle Tracking and Viability Assessment

For ELV uptake in vitro evaluation, Alexa Fluor 555-labelled particle solution containing 0.5 mg/mL of total protein in PBS was applied on mixed glial and microglial cultures. After 20 min of incubation, Hoechst33342 (6 µg/mL, Thermo Fisher Scientific, Vilnius, Lithuania) was added to the incubation medium for visualisation of nuclei and isolectin GS-IB_4_ from *Griffonia simplificolia*, Alexa Fluor^®^ 488 conjugate (10 ng/mL, Molecular Probes^TM^, Fisher Scientific, Vilnius, Lithuania) for staining microglial with microglial and mixed glial using fluorescence microscope Zeiss Axio Observer.Z1 (Carl Zeiss, Jena, Germany).

The viability of microglial cells in pure microglial cultures after ELV uptake was assessed by double nuclear staining with fluorescent dyes Hoechst33342 (6 µg/mL) and propidium iodide (3 µg/mL) for 10 min and imaging in a fluorescent microscope Olympus IX71 (Olympus Corporation, Tokyo, Japan). The images were taken by a 01-Exi-AQA-R-F-M-14-C camera (QImaging, Surrey, BC, Canada) and the image analysis was performed by the ImageJ software.

### 2.9. Immunohistochemistry of Brain Tissue

Animals were sacrificed by cervical dislocation; brains were removed, divided into two equal pieces along the longitudinal fissure. One part of each brain was snap-frozen in liquid nitrogen and stored for qPCR assay. Another part was washed in PBS and embedded in 4% paraformaldehyde solution for 30 min. Afterwards, the brains were kept in 25% sucrose for 24 h at 4 °C. Then, the tissue was frozen in liquid nitrogen and stored at −80 °C until further processing. Next, serial coronal sections (20 μm thick) containing the substantia nigra were cut at −23 °C using a cryostat (HM 560 Microm, Walldorf, Germany). The sections were mounted on glass slides (Plus, Menzel Glaser, Thermo Fisher Scientific, Vilnius, Lithuania), and allowed for complete dehydration in the cryostat chamber for 10 min. Next, the slides were washed with PBS solution (3 × 10 min) and stained with 0.5 mM 4′,6-diamidino-2-phenylindole (DAPI) for 5 min at room temperature for visualisation of the nuclei.

For microglial cell staining, the sections were subjected to 0.5% Triton X-100 permeabilisation for 40–60 min at room temperature in a dark, humid environment. After+ washing with PBS (3 × 10 min), the sections were incubated with 1 µg/mL primary rabbit monoclonal antibodies against microglial transmembrane protein TMEM119 (RRID:AB_2800343, ab209064, Abcam) overnight at 4 °C and with AlexaFluor^®^ 488 conjugated chicken anti-rabbit IgG (Thermo Fisher Scientific, Vilnius, Lithuania) secondary antibodies diluted in PBS 1:200 for 2 h at room temperature. The slides were coated with anti-fading oil (Vectashield, Vector Laboratories, Burlingame, CA, USA), the edges were varnished with colourless nail polish. For negative control, PBS solution was added instead of primary antibodies, followed by secondary antibodies, and there was no fluorescence observed in the negative control samples. The stained tissue was visualised by laser scanning confocal microscope: Zeiss Axio Observer LSM 700 (Carl Zeiss Microimaging Inc., Jena, Germany).

The fluorescence intensity of AF555 signal and colocalisation analysis in brain slice micrographs was performed by ImageJ freeware. For each evaluation group, 12 micrographs of 320 μm × 320 μm from 3 separate samples were assessed for the average signal strength intensity between the minimal—maximal values of 0 and 250 relative fluorescent units (RFU), respectively. The data are presented as averages ± standard deviation.

### 2.10. Evaluation of Intracellular and Intramitochondrial ROS

Cytoplasmic and intramitochondrial ROS were evaluated as described in [28]. Briefly, the 2′, 7′-dichlorofluorescein diacetate (DCFDA, Invitrogen, Thermo Fisher Scientific, Vilnius, Lithuania) was used to assess the formation of intracellular ROS in microglia. Microglial cells were seeded in 96-well plates (50,000/well) and cultivated for 24 h. After incubation, cells were treated with poly (I:C)-primed and not primed airway cell ELVs (10 μg/mL of total protein) for 16–18 h, or with poly (I:C) for 1 h. After incubation, all cells were washed with Hank’s Balanced Salt Solution (HBSS, Gibco^TM^, Thermo Fisher Scientific, Vilnius, Lithuania). Following washes, cells were stained with 10 μM DCFDA for 30 min at 37 °C, repeatedly washed with HBSS and visualised using Olympus IX2-ILL100 fluorescence microscope (Olympus, Hamburg, Germany). For evaluation of mitochondrial superoxide, the cells were grown in clear 96-well plates the same way as for cytoplasmic ROS assessment. The cells were 3x washed with HBSS and incubated with 2 µM MitoSox^TM^ Red (Thermo Fisher Scientific, Vilnius, Lithuania) in HBSS at 37 °C in the dark for 15 min. For positive control, the cells were treated with 100 μM Antimycin A for 30 min. The images were taken by fluorescent microscope Zeiss Axio Observer.Z1, and fluorescence intensity was evaluated by ImageJ software (National Institute of Health, Bethesda, MD, USA).

### 2.11. Real-Time Quantitative Reverse Transcription-PCR

Gene expression level of antiviral response pathway cytokines was evaluated in mouse brains 24 h after intranasal delivery of poly (I:C)-primed airway ELVs and in cultured microglial cells after 24-h incubation with the same ELVs. Total RNA was isolated using TRIzol™ reagent (Invitrogen, Thermo Fisher Scientific, Vilnius, Lithuania) according to the manufacturer’s instructions. High-Capacity cDNA Reverse Transcription Kit (Applied Biosystems™, Thermo Fisher Scientific, Bleiswijk, The Netherlands) was used for cDNA synthesis after DNase (Thermo Fisher Scientific, Vilnius, Lithuania) treatment. Quantitative reverse-transcription PCR (qRT-PCR) applying SYBR Green I assay was used to analyse mRNA expression of inflammation markers: interleukin-6 (*Il6*), interferon beta-1 (*Ifnb1*), prostaglandin-endoperoxide synthase-2 (*Ptgs2*), chemokine (C-C motif) ligand 5 (*Ccl5*), interferon-alpha (*Ifna*), interleukin-1-beta (*Il1b*), tumour necrosis factor-alpha (*Tnf*), interferon-gamma (*Ifng*). Actin beta (*Actb*), glyceraldehyde 3-phosphate dehydrogenase (*Gapdh*) and RNA Polymerase II Subunit A (*Polr2a*) genes were used as endogenous controls for signal normalisation. PCR was carried out in a total volume of 12 µL and consisted of 6 µL of Power SYBR™ Green PCR Master Mix (Applied Biosystems™, Thermo Fisher Scientific, Bleiswijk, The Netherlands), 15 ng of cDNA, and 0.25 µM of each primer and nuclease-free water. Real-Time PCR System “Applied Biosystems 7500 Fast” (Applied Biosystems™, Thermo Fisher Scientific, Bleiswijk, The Netherlands) used for products amplification and fluorescent signal registration. Comparative 2^−ΔΔCT^ method was used to evaluate inflammation markers expression in poly (I:C)-primed airway ELVs-treated samples compared to those treated with not primed ELVs.

### 2.12. Statistical Analysis

Statistical analysis was performed using Sigma Plot v12.5 software (Systat Software Inc., Berkshire, UK). The means of the experimental data are presented with standard errors. Statistical comparisons of the two groups were performed using Student’s *t*-test. Multiple groups were compared using and one-way analysis of variance (ONE WAY ANOVA) with Bonferroni statistical criterion. Differences between means were considered statistically significant at *p* < 0.05.

## 3. Results

### 3.1. Identification and Characterisation of Airway Cell ELVs

At first, ELVs isolated from poly (I:C)-primed and not primed rat airway cell-conditioned medium were evaluated for morphology by transmission electron microscopy (TEM). The samples contained vesicles of approximately 10–190 nm in diameter (Figure 2a,d).

The dynamic light scattering analysis revealed that the particle diameter in samples ranged from 10 to 160 nm, and 98 per cent of the particles in the samples were in the range of 30–150 nm, which is characteristic for exosomes (Figure 2b,e). Next, particle preparations were ELISA-tested for common exosomal markers, tetraspanins CD9 and CD63. The analysis confirmed both markers present in the samples; however, CD63 was found about five times more than CD9 (Figure 2c). Similar characteristics were found by examining ELV preparations from mice airway cell cultures; however, in contrast to the rat samples, CD9 marker predominated (Figure 2f). There were no detectable differences in morphology, size distribution and CD63/CD9 presence found between poly (I:C)-primed and not primed airway cell exosomes (data not shown).

### 3.2. Poly (I:C)-Primed Airway ELV Tracking in the Brain

For blood-brain barrier penetration monitoring, Alexa Fluor-555 (AF555) labelled ELVs from poly (I:C)-primed and not primed airway cells were introduced intranasally to mice. Red fluorescence became visible in the brain (middle section of coronal plane slices) already 1 h after treatment, and later (after 3 and 5 h), the fluorescence spots became more extensive (Figure 3a). Some of them resembled cell body shape suggesting that particles were internalised by specific cells at this time point. Interestingly, some brain regions, such as the hippocampal pyramidal neuron layer, were more prone to collect ELVs (Figure 3b). In the slices from the olfactory bulb (Figure 3c), it was no specific particle clustering, but the cell-shaped AF555-positive staining was present already after 1 h of the ELV delivery. We did not detect visual differences between particle distribution in the brains treated with poly (I:C)-primed and not primed airway ELVs. Additionally, there was no red staining visible in brain slices from mice that did not receive ELV treatment (0 h image in Figure 3a).

The quantitative evaluation of AF555 fluorescence intensity after 1, 3 and 5 h revealed a significant signal increase from about 1 relative fluorescent unit (RFU) at time point “zero” up to about 5 RFUs after 1 h and further grew to about 7 RFUs after 3 h (Figure 3d). However, there was no significant AF555 fluorescence signal increase in the ELV treated brain samples after 5 h compared to those after 3 h. Additionally, no significant differences between brains treated with poly (I:C)-primed and unprimed airway cell ELVs at each time point.

Next, the brain slices were immunostained for TMEM19 to determine if the ELVs were internalised by microglial cells. The staining revealed that most particles colocalised with microglial cell bodies (Figure 4a, upper image row) or their processes (zoomed-in ROI in the lower image row of Figure 4a).

The colocalisation analysis of the ROI marked by the white line in the zoomed (lower) image row revealed a strong correlation between ELV and microglial staining (Figure 4b). The diagonal positioning of cytofluorograms and close to 1 Pearson’s correlation coefficient (PCC) indicates a similar distribution of red (ELVs) and green (microglia) pixel intensity. Mander’s 1 coefficient (M1) value is 1.00, and this means that all the AF555-positive area completely overlaps with the AF488-positive area. Similarly, the values of M1 were found close to 1 in all examined images of brain slices after intranasal delivery of the ELVs (Figure 4c). The values of PCC in the full-size images were not as high as in the ROI analysis, and this can be explained by the considerable difference between the red and green fluorescence areas.

Overall, the immunohistochemistry data indicate that airway cell ELVs, after intranasal delivery, enter the brain within hours and are actively internalised by microglial cells.

### 3.3. Poly (I:C)-Primed Airway ELV Tracking in Glial Cell Cultures

In vitro experiments of internalisation of Alexa Fluor-555 (AF555) labelled ELVs from poly (I:C)-primed and not primed airway cells in cultivated rat pure microglial and mixed glial cultures confirmed that microglial cells internalise particles more rapidly than astrocytes. All particles were entirely relocated to microglial cells 30 min after addition to the culture medium in both pure microglial and mixed glial cultures (Figure 5). The ELVs looked as if they collected into cytoplasmic vesicles suggesting that the uptake pathway was either endocytosis, phagocytosis, or micropinocytosis. Such particle internalisation did not affect the viability of microglial cells; double nuclear fluorescent staining with Hoechst 33342 and propidium iodide revealed no significant difference in viable cell number between ELV-treated and untreated, as well as between poly (I:C)-primed and not primed airway ELV-treated microglial cultures; the percentage of viable cells in all cultures were above 98% (Figure A3).

### 3.4. Poly (I:C)-Primed Airway Cell ELVs Impact on ROS Formation in Microglia

Mitochondrial and cytoplasmic ROS act upstream of the inflammatory cascade during viral infection and play a crucial role in immune memory formation [29,30,31]. To determine whether poly (I:C)-primed airway ELVs affect mitochondrial and cytoplasmic ROS production, microglial cells were incubated with the ELVs for 16–18 h, loaded with either MitoSOX^TM^ or DCFDA and monitored under a fluorescent microscope.

Both visual monitoring and quantitative evaluation of the micrographs revealed that poly (I:C)-stimulated airway cell ELVs significantly increased the formation of mitochondrial ROS in microglial cells (Figure 6a). After 24 h, the MitoSOX^TM^ fluorescence intensity was by 31% higher compared to the samples treated with untreated airway cell ELVs. Moreover, the increased level of mitochondrial ROS in poly (I:C)-primed airway cell ELV-treated samples remained even after 48 h after the treatment. Additionally, ELVs from poly (I:C)-primed airway cells significantly increased the formation of cytoplasmic ROS in microglia; the DCFDA fluorescence level after 24 h post-treatment rose by 14% (Figure 6b). However, there was no significant difference in MitoSOX^TM^ and DCFDA fluorescence intensity between untreated control (100%) and samples treated with ELVs from airway cells unaffected by poly (I:C).

### 3.5. Antiviral Inflammatory Response Related Cytokine Expression in Brain and Cultured Microglia after Treatment with Poly (I:C)-Primed Airway Cell ELVs

An increase in ROS production by poly (I:C)-primed airway ELVs in cultured microglial cells suggested they could be inflammatory activated. Therefore, the next step in the study was to test whether virus mimetic poly (I:C)-treated ELVs might stimulate expression of inflammatory cytokines involved in the antiviral inflammatory response in brain tissue and microglial cell cultures. The next day after in vivo intranasal treatment with poly (I:C)-primed airway ELVs, the average level of mRNA of *Ifna*, *Ifng*, *Ccl5*, *Il1b*, *Tnf*, and *Ptgs2* tended to increase compared to the level in brains treated with not primed airway ELVs and reaching statistically significant difference for *Ptgs2* (Figure 7a). A similar tendency was observed in microglial cultures after 24-h treatment with poly (I:C)-primed airway ELVs, with a statistically significant increase in *Ccl5* mRNA level (Figure 7b).

To summarise, ELVs derived from airway cells affected by virus mimicking poly (I:C) sequence tend to elevate antiviral response cytokine expression both in the brain after intranasal delivery and in cultured microglia after cell culture medium.

## 4. Discussion

The main highlights of the present study include confirmation that (1) airway cell ELVs easily penetrate the blood-brain barrier of healthy laboratory mice and (2) are quickly (within an hour or two) internalised by microglial cells. Moreover, (3) the ELVs from airway cells after viral infection mimicking priming induce specific changes in microglial cells, leading to increased intracellular and intramitochondrial ROS generation. Finally, (4) the ELVs stimulate inflammatory cytokine expression in the brain and microglia.

Upper respiratory tract infections result in 10 million outpatient visits per year, and 70–90% of these infections are of viral origin [32,33]. Viral infections promote innate or nonspecific and acquired or specific immune responses. There is a well-established correlation between peripheral viral infection and neuroinflammation, leading to neurodegeneration [34,35]. The recently emerged SARS-CoV-2 virus that caused COVID-19 pandemic is also not an exception from this point of view; there are numerous reports about neuroinflammation-related central nervous system damage caused by this infection [36]. However, the mechanisms transmitting inflammation from the periphery to the brain remains elusive.

One of the most suggestable candidates for inflammation transmission from the periphery to the brain is exosomes. Exosomes of cells affected by bacteria, viruses, parasites, or fungi carry pathogen components that can be transferred to other cells [37]. For example, exosomes produced by bacterial infection-affected macrophages contain pro-inflammatory factors that activate B and T lymphocytes enhancing the immune response [38,39]. Virus-treated cells contain viral proteins and RNA that cause inflammatory response or even infection in recipient cells [37]. Herpes simplex, hepatitis A, B, C, and human immunodeficiency (HIV) viruses can spread through exosomes [40,41,42,43]. The above-listed evidence allows assuming that some of the exosomes produced by infected cells could penetrate the central nervous system and cause neuroinflammation. Although several studies report peripheral exosomes crossing the blood-brain barrier [41,42], their further cellular uptake remained unknown. In addition, none of the examined exosomes originated from airway epithelium or fibroblasts. Our study experimentally demonstrated fast brain uptake of airway epithelial ELVs and provided evidence about their localisation in microglial cells. In addition, the in vivo data were supported by in vitro experimental proof of much faster ELV uptake performance by microglia compared to astrocytes. Although this is not direct evidence of airway infection spreading to the brain, the study gives a solid reason and background for studying this possibility. Interestingly, there were visually more airway cell ELVs in the pyramidal neuron area of the hippocampus after intranasal introduction. Hippocampal neurons are the primary target of Alzheimer’s disease, and the cause of such selective damage to this area remains unknown. The fact that the hippocampus selectively collects peripheral ELVs carrying inflammatory mediators suggests a new hypothesis for Alzheimer’s disease development. The hippocampus and olfactory regions are anatomically close; they are connected by memory formation networks and participate in glymphatic brain clearance [44,45]. Such anatomical and functional relations likely favour ELVs transmission from the nasal cavity. Similar to our findings, microglial localisation of intravenously injected fluorescently labelled exosomes was observed by Li and colleagues [46]. The authors have found that exosomes from serum of LPS-treated mice initiate inflammation in the brains of healthy exosome recipients. Additionally, an in vivo study by Zhuang with co-authors demonstrates fast uptake of exosomes (but not microvesicles) from three different cancer cell lines and embryonic fibroblasts by brain and microglia [47].

Despite showing ELV relocation from the nasal cavity to the brain cells, we did not progress the understanding of the mechanism of EV entrance to the CNS. A recent review by Saint-Pol and colleagues suggest at least five possible theoretical interactions of the peripheral exosomes with the blood-brain barrier forming cells: (i) association with a protein G-coupled receptor on the cell surface, (ii) adhesion to the cell surface and fusion, releasing the EV content in the cytoplasm, (iii) micropinocytosis, (iii) nonspecific/lipid raft formation, and (iv) receptor-mediated transcytosis [8]. Banks and co-authors discovered that intravenously introduced exosomes from mouse, human, cancerous, and non-cancerous cell lines all cross the blood-brain barrier, but at different rates and by distinct mechanisms: adsorptive transcytosis and mannose 6-phosphate receptor [6]. Furthermore, both in vivo and in vitro experiments confirm a substantial increase in the barrier permeability for exosomes in the presence of inflammatory stimuli such as bacterial LPS or TNFα [6,48]. Such findings strongly suggest that peripheral EVs might be important yet underestimated players in neuroinflammation.

Exosome researchers inevitably have to deal with certain limitations of the investigation tools, and this study was not an exception. A significant limitation is the lack of reliable and straightforward isolation techniques to collect enough exosomes for investigation and avoid impurities such as proteins, lipid particles and other extracellular vesicles [49]. Usually, a relatively pure sample means low gain and vice versa [50,51]. In addition, the characteristics of the exosome population, such as dominant particle size, biomarker content and biological efficiency, might vary depending on chosen isolation procedure [52,53]. The polymer precipitation method of EV isolation applied in this study for simplicity and gain is characterised as a high-yield-low-purity method. This might raise the possibility that some part of the biological activity of the preparation might be attributed to the non-exosomal contaminants in the preparation. Another important limitation is a lack of specific exosome identification procedures. There is a common practice to identify exosomes by size in a nanoparticle tracking analyser, by morphology (electron microscopy, usually transmission electron microscopy that allows visualising lipid bilayer of the exosomal membranes) and by specific markers. However, even if each of the steps confirms exosome-like qualities, there is still possible that some particles in the preparation are of different origin than exosomes [54,55]. Therefore, the hypothesis of the study would be suggested to reconfirm after a while when more specific markers and high yield-high-purity exosome isolation assays are elaborated.

This study demonstrated that ELVs from virus mimetic-treated airway cells stimulate ROS generation in cultivated microglia. We have applied two different assays for ROS evaluation. Conversion of DCFDA to fluorescent 2,7-dichlorofluorescein (DCF) indicates hydroxyl, peroxyl and some other species produced into cell cytoplasm [56]. Mitochondrially targeted MitoSOX^TM^ Red, in its turn, reports about the intensity of generation of superoxide radicals within mitochondria. Of all ROS, the respiratory chain primarily produces superoxide, which cannot cross membranes and remains where made unless converted to hydrogen peroxide by superoxide dismutase. Hydrogen peroxide molecules penetrate mitochondrial membranes to the cytosol, where it often undergoes further conversions depending on the chemical environment, e.g., to hydroxyl radical by Fenton reaction. Thus, mitochondrial superoxide could also be the source of cytoplasmic ROS detected by the DCFDA reaction. The fact that poly (I:C)-primed airway ELVs stimulated mitochondrial superoxide generation far more intensively than intracellular (cytoplasmic) ROS strongly suggests mitochondria were the primary source of radicals in this case. However, more experimental evidence is required to prove this hypothesis, such as testing how microglial cytoplasmic ROS levels induced by poly (I:C)-primed airway ELVs would be affected by the presence of mitochondrial superoxide scavenger mitoTEMPO.

Reactive oxygen species, including those from mitochondria, are generated as a fast response to infection [29,57]. Mitochondrial ROS lie at the top of the innate immune response cascade and precedes ROS generation from phagocytosis-related NADPH oxidase [58]. Moreover, they are required for virus-induced mitochondrial antiviral signalling protein (MAVS) to activate inflammasome [59,60], which is directly linked to neurodegeneration according to recent research [61,62]. Such an important role of mitochondrial ROS in inflammatory response indicate the following landmarks in dissecting the role of virus-primed airway cell ELVs, such as examining their effect on the mitochondrial and glycolytic profile of microglial cells, inflammasome activation and impact on the inflammatory cytokine level in the brain.

The critical finding of the study is the increased level of expression of antiviral inflammatory response marker CCL5, or Rantes, in the brain. Interferons alpha and gamma belong to different interferon groups, and both are known to activate the NFkappaB pathway, which stimulates CCL5, IL-1β and TNF-α [63,64]. Thus, the results suggest that at least one of the pathways might be stimulated by poly (I:C)-primed airway ELVs. The type I pathway is well known to be directly activated by the TLR3 [65]. However, there is a report that this receptor might also trigger the interferon type II pathway [66]. Such evidence suggests that ELVs from poly (I:C)-treated airway cells carry some TLR3 agonist, possibly, poly (I:C) itself. Indeed, poly (I:C) was transferred in extracellular vesicles of U937 macrophages, and such vesicles mimicked the direct effect of poly (I:C) on synovial fibroblasts [67]. Therefore, it is very likely that poly (I:C) was also among the cargo of the airway cell ELVs examined in this study. Interestingly, a transcription of a gene involved in cyclooxygenase-prostaglandin inflammatory signalling PTGS2 was significantly activated in cultured microglia after poly (I:C)-primed airway ELV treatment. The PTGS2 pathway modulates the extend of antiviral response and might play a significant role in autoimmune disease development [68]. Of note, this enzyme plays an essential role in Parkinson’s and Alzheimer’s disease-related neurodegeneration [69,70]. Overall, the elevated levels of cytokine expression after treatment with ELVs from virus mimetic-primed airway cells provides new hallmarks for investigation of peripheral-central nervous system vesicular communication, such as determining the exact signalling sequence and actual protein levels, detecting differences between such signalling induced by different viruses, and examining correlations with neurodegenerative signalling pathways. Additionally, elucidating the mechanism of ELV crossing the blood-brain barrier under the ordinary and infection-affected conditions and identifying the virus-induced molecular profile of airway cell ELVs would be of great importance in understanding the role of these extracellular vesicles in communication between the periphery and brain during viral infections.

## 5. Conclusions

Virus mimetic poly (I:C)-primed and not primed airway cell ELVs reach brain tissue after not more than an hour from the intranasal introduction in mice.Both in the brain and culture, airway cell ELVs are internalised by microglial cells faster than by other cell types, such as astrocytes.Poly (I:C)-primed airway exosomes induce a significant and lasting increase in cytoplasmic and intramitochondrial ROS production. Conversely, the exosomes from not primed airway cells do not cause changes in ROS levels.Poly (I:C)-stimulated airway exosomes significantly stimulate expression of inflammatory factors *Ccl5* (in brain) and *Ptgs2* (in cultured microglia).

## Figures and Tables

**Figure 1 biology-10-01359-f001:**
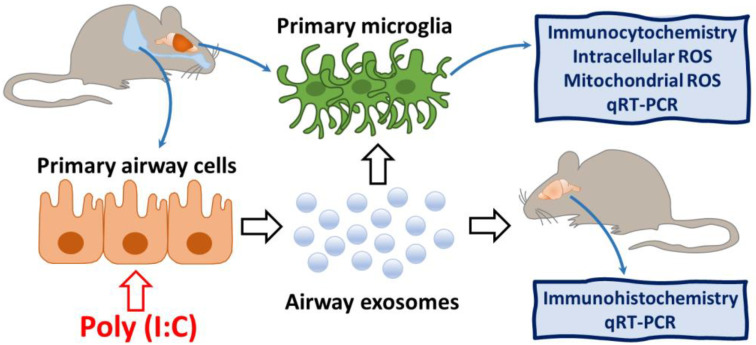
The experimental design of the study. Primary airway cells of rodent origin were treated with poly (I:C), and their ELVs were applied on microglial cell cultures and intranasally to mice. Microglial cells in vitro and in vivo were investigated for particle internalisation, and cultured microglia were analysed for intracellular and intramitochondrial ROS production.

**Figure 2 biology-10-01359-f002:**
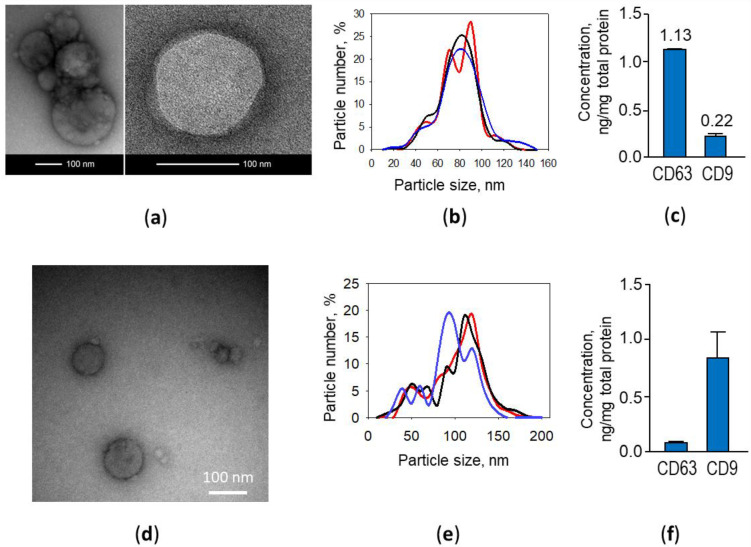
Poly (I:C)-treated rat (**a**–**c**) and mouse (**d**–**f**) airway cell ELV morphology, particle size distribution and specific markers. Representative transmission electron microscopy images (**a**,**d**), dynamic light scattering nanoparticle analysis data of three exosome samples (**b**,**e**) and tetraspanin CD63 and CD9 content per total exosome preparation protein determined by ELISA (**c**,**f**).

**Figure 3 biology-10-01359-f003:**
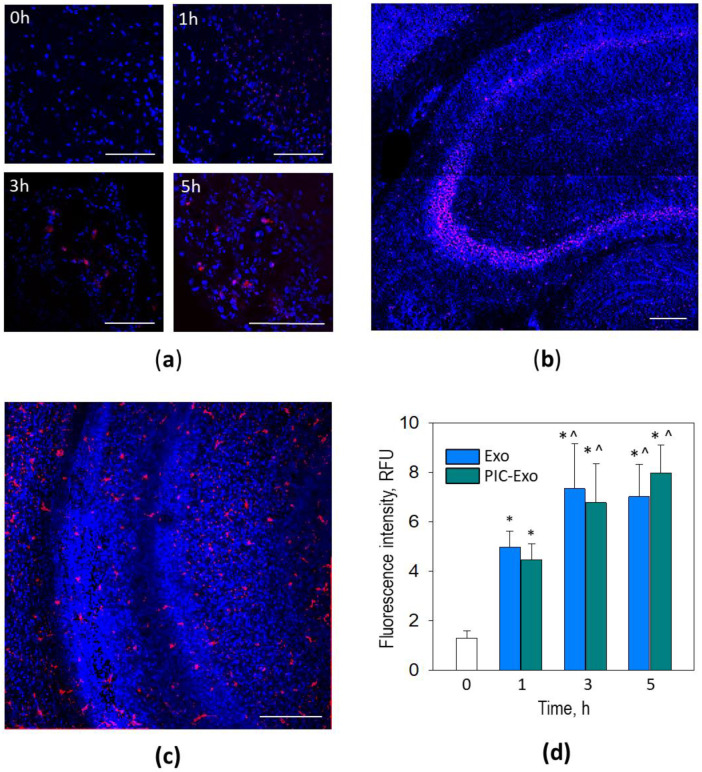
ELVs from poly (I:C)-treated airway cells in mouse brain coronal slices from central section after 0, 1, 3, and 5 h following intranasal introduction (**a**). In (**b**)—quantitative evaluation of—AFF555 fluorescence intensity in brain slice micrographs 3 h after intranasal administration of the stained particles. The data are expressed as averages ± standard deviation of 3 independent experiments that involved three animals in each experimental group; the fluorescence intensity was evaluated in 12 images for each separate animal sample. PIC-Exo is for poly (I:C)-primed airway ELVs, Exo—not primed airway ELVs, and controls brains from mice without treatment. * indicates statistically significant difference compared to the Control when *p* < 0.001, ^—compared to Exo or PIC-Exo, respectively, after one hour, when *p* < 0.05. In (**c**,**d**), there are ELVs in the olfactory bulb, prefrontal cortex and hippocampus slices, respectively, one hour after intranasal introduction. Particles loaded with RNR-conjugated AF555 are red, and nuclei are stained blue with DAPI. Scale bar 100 μm.

**Figure 4 biology-10-01359-f004:**
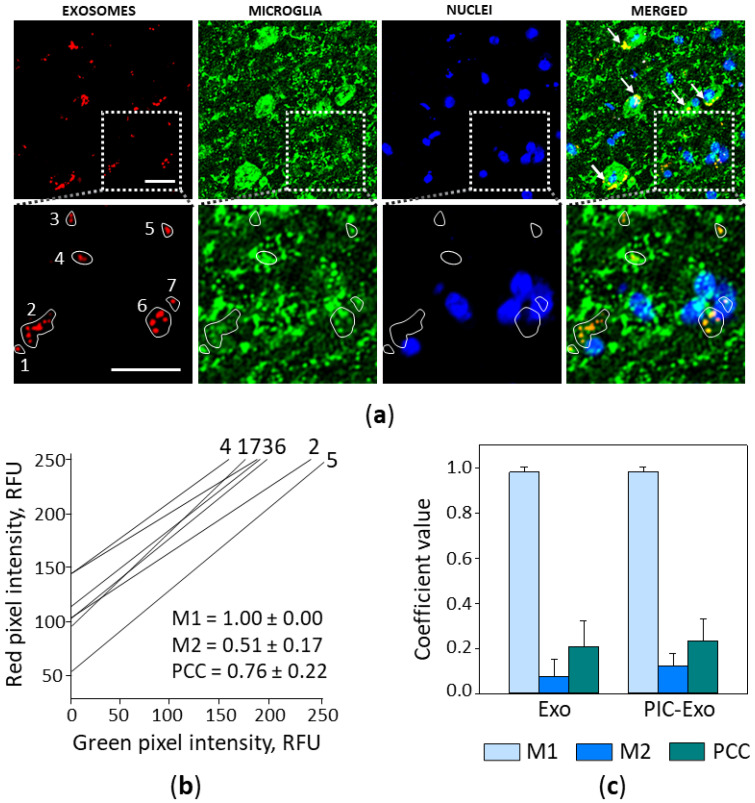
Poly (I:C)-treated airway ELV internalisation by microglia in mouse brain 2 h after intranasal introduction. In (**a**), a representative brain slice image, where particles loaded with RNR-conjugated AF555 are red, nuclei are stained blue with DAPI, and microglial cells are green, visualised by immunostaining against TMEM119. White arrows indicate microglial cell bodies with colocalised ELVs (upper image row, MERGED), and particles colocalising with microglial processes is visible in the zoomed ROI image (lower image row). The scale bar is 20 μm. Colocalisation analysis in seven small ROIs indicated by the white lines was performed by ImageJ plugin JACoP; cytofluorograms, Mander’s and Pearson’s coefficient values are presented in (**b**). Results of colocalisation analysis in full-size images are shown in (**c**), n = 15. M1—Mander’s overlapping coefficient showing the red fluorescence fraction overlapping the green, M1—the green fraction overlapping the red; PCC—Pearson’s correlation coefficient. Exo—images of the slices from brains treated with ELVs from control airway cells; PIC-Exo—from poly (I:C)-treated airway cells.

**Figure 5 biology-10-01359-f005:**
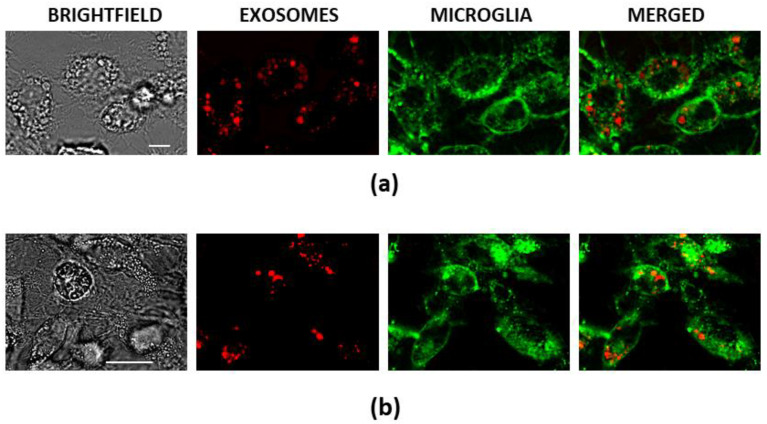
Poly (I:C)-treated airway cell ELV internalisation by microglia in and in pure microglial (**a**) and mixed rat (**b**) glial cultures 30 min after treatment. Particles loaded with RNR-conjugated AF555 are red, nuclei are stained blue with Hoechst33342, and microglial cells are green, stained with AF488-conjugated isolectin B4. The scale bar in (**a**) is 10 μm and in (**b**)—50 μm.

**Figure 6 biology-10-01359-f006:**
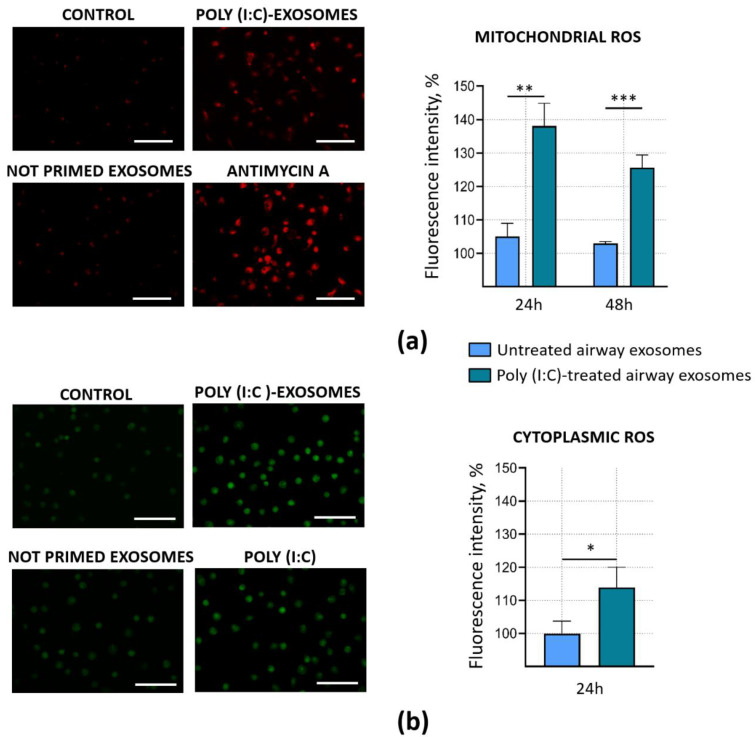
Intramitochondrial and cytoplasmic reactive oxygen species (ROS) formation in microglia after treatment with ELVs from poly (I:C)-primed airway cells. Cytoplasmic ROS were detected by DCFDA assay and mitochondrial superoxide—by MitoSOX^TM^ fluorescence. (**a**)—representative images and quantitative evaluation of MitoSOX fluorescence in cultured microglia. For positive control of the assay, 100 μM Antimycin A was used. The scale bar is 100 μm. (**b**)—representative images and quantitative evaluation of 2,7-dichlorofluorescein (derived from DCFDA) fluorescence in microglial cultures. For positive control of the assay, 1 μg/mL poly (I:C) was used. The scale bar is 100 μm. The quantitative data of ROS-dependent fluorescence intensity in micrographs are presented as percentage of untreated control and expressed as averages ± standard deviation of 3 experiments in 3 biological replicates. *** indicates statistically significant difference when *p* < 0.001; **—*p* < 0.01; *—*p* < 0.05.

**Figure 7 biology-10-01359-f007:**
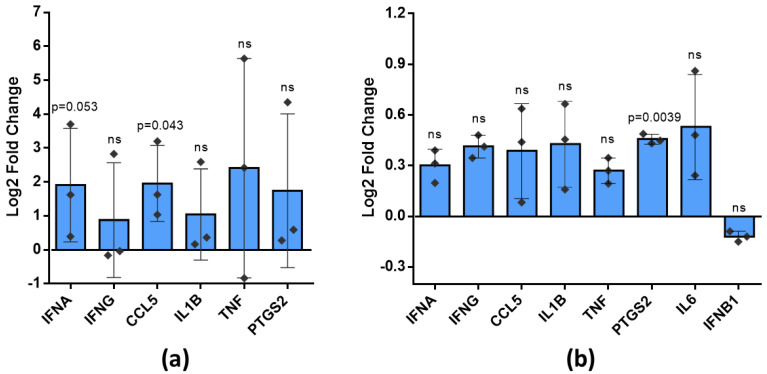
Expression of mRNA of inflammation markers after poly (I:C)-primed airway ELV treatment in (**a**) brain tissue and (**b**) cultured rat microglial cells. Data are given as log2 Fold Change (2^−ΔΔCT^) calculated from control specimens treated with unprimed ELVs. Statistical differences of markers expression between control specimens and specimens treated with poly (I:C)-primed airway ELVs calculated applying unpaired *t*-test; *p* < 0.05 considered significant. Bar plot represent mean of 3 experiments and whiskers—standard deviation; ns—not significant. *Ifna* is for interferon-alpha gene, *Ifng*—interferon-gamma, *Ccl5*—chemokine (C-C motif) ligand 5, or Rantes, *Il1b*—interleukin-1-beta, *Tnf*—tumour necrosis factor-alpha, *Ptgs2*—prostaglandin-endoperoxide synthase-2, *Il6*—interleukin-6, *Ifnb1*—interferon beta-1.

## Data Availability

The raw data supporting the conclusions of this manuscript will be made available by the authors, without undue reservation, to any qualified researcher.

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
