# Peer review of "Virus Mimetic Poly (I:C)-Primed Airway Exosome-like Particles Enter Brain and Induce Inflammatory Cytokines and Mitochondrial Reactive Oxygen Species in Microglia"

_biology, 2021, doi:10.3390/biology10121359_

Round 1

Reviewer 1 Report

In the present study, the authors have investigated the transfer of the airway exosomes from the respiratory tract to the brain and how these exosomes affect microglia and stimulate immunity responses in the brain microenvironment. Taken together, this is an interesting, novel and important study, however, there are a few minor points that need to be addressed by the authors:

1-Hours should be written as h throughout the manuscript, days as d, etc.

2-line 231, please correct 0,5 mg to 0.5 mg

3-line 246, please specify whether it is qRT-PCR or qPCR. Please mention the full name for qRT-PCR, quantitative reverse transcription-PCR (qRT-PCR) at its first occurrence in the text.

4-No need to mention the cat number for Trizol, etc. The materials and methods could be more concise to make the manuscript shorter.

5-line 122: The correct title is primary culture of airway cells.

6-Murine genes should be italicized, with only the first letter in upper-case.

Author Response

Response to Reviewer 1

We thank the Reviewer for revising our study and for the improving comments.

1.Hours should be written as h throughout the manuscript, days as d, etc.

We have carefully corrected abbreviations according to the comment.

  1. Line 231, please correct 0,5 mg to 0.5 mg

We thank the Reviewer for the observation; it has been corrected.

  1. Line 246, please specify whether it is qRT-PCR or qPCR. Please mention the full name for qRT-PCR, quantitative reverse transcription-PCR (qRT-PCR) at its first occurrence in the text.

We have specified and corrected abbreviations according to the comment.

  1. No need to mention the cat number for Trizol, etc. The materials and methods could be more concise to make the manuscript shorter.

We have deleted the catalogue numbers and shortened the description of some methods.

  1. Line 122: The correct title is primary culture of airway cells.

The title is corrected accordingly.

  1. Murine genes should be italicized, with only the first letter in upper-case.

We thank you for the remark; we have corrected the gene indicators accordingly.

Reviewer 2 Report

The authors proposed a paper focused on the role of viral like-modified airway exosomes on brain in promoting a microglia-mediated inflammatory response.

The scientific soundness of this study would be greater after some major and crucial modifications. Please see below my recommendations:

  • It seems that exosomes, or small extracellular vesicles (EVs), can cross the blood-brain barrier through unknown or understudied pathways (reviewed in Saint-Pol et al., 2020, Cells). The technical approach used here can suggest this entry through the BBB, however without any control. Please reconsider to study the transport of an other compound as a control to prove it, and also to check the BBB integrity in the in vivo mouse model used. Classical markers used are labelled Dextrans or small lipophilic compounds.
  • Introduction:
    • A/some reference/s to explain the possible entry of EVs through the BBB would be appreciable. 
    • Some recent studies or reviews can be cited to define exosomes, especially since 'exosome' is a subgroup of the small EVs fraction (Saint-Pol et al., 2020 Cells; Kowal et al., 2016 PNAS; Mathieu et al., 2019 Nat Cell Biol; 2021 Nat Comm; Théry et al., 2018 JEV). Please be careful also all along the manuscript regarding that point.
  • Results:
    • Figure 2 and A1 can be melted. 
    • Concerning the 'exosome' characterization, morphology and size fit with small EVs, but only 2 tetraspanins as 'positive markers' is clearly not enough. Please include other 'positive markers' as proves of their endosomal origin (syntenin-1, Tsg101) and some 'negative markers' to complete your demonstration, such as calnexin, HSP70. Without these complementary experiments, the name 'exosome' is excluded. Please see Thery et al., 2018 (JEV) to follow the updated Minimal Instructions for the study of Extracellular Vesicles (MISEV).
  • Conclusions need to be measured, and not affirmed since the demonstration is slight for some points.
  • Reference 7 is not well formatted in the list.

Author Response

Response to Reviewer 2

We thank the Reviewer for the careful revision of our study and for the improving comments.

  1. It seems that exosomes, or small extracellular vesicles (EVs), can cross the blood-brain barrier through unknown or understudied pathways (reviewed in Saint-Pol et al., 2020, Cells). The technical approach used here can suggest this entry through the BBB, however without any control. Please reconsider to study the transport of an other compound as a control to prove it, and also to check the BBB integrity in the in vivo mouse model used. Classical markers used are labelled Dextrans or small lipophilic compounds.

We agree that the elucidation of the pathway by which exosomes may pass from the periphery to the brain is crucial in understanding exosome biology . Therefore, we have extended the Discussion, including some findings and theoretical suggestions regarding  EV crossing the BBB, also citing the review suggested (lines 519-532). On the other hand, our study was not focused on the mechanism of crossing the BBB but rather on defining the possible impact of the EVs after their translocation to the brain. We do not entirely agree that the intranasal introduction of the EVs together with the classical BBB integrity tracers would help to reveal the mechanism of EV penetration. First of all, the fluorescent dextrans are not concentrated in cells enough to be visualised under a microscope; thus, it would not give any clue about the similarities or differences in the mechanism of brain entrance. Even if such visualisation would be possible, it could not directly confirm the same (or different) entrance pathway. We think that for the pathway studies, the impact of distinct cell recognition and transport pathway elements should be examined, similarly as in the study by Zhuang et al. [1].

However, we did examine the BBB integrity of Balb/c mice that were used for the intranasal EV introduction study. Thirty minutes after intraperitoneal injection of 200 mL solution of 2 mM Micro Ruby 3000 Da and Blue Dextran 2000 kDa (performed as described in [2]), and BBB-impermeable anticancer drug doxorubicin (DOX) administered at 0,142 mg per mouse (47 mL solution of 3 mg/mL), in the centrifuged brain homogenates, prepared immediately in PBS for the dextran tracers, and in 50 % DMSO/PBS (v/v) for DOX, the average 3 sample concentration of the 3000 Da dextran was found 455.25±11.40 mM (ex/em wavelengths 555/580 nm, concentration calculated according to the calibration equation). There was no increase in the fluorescence in the brains of the control mice that were administered PBS solution without the tracers. Also, there was no DOX fluorescence (ex/em wavelengths 485/595 nm) or 620 nm light absorption by Blue Dextran 2000 detected in brain homogenates of both treated and untreated mice. The experiment indicates that the integrity of the BBB in the mice was not impaired.

  1. Zhuang, X.; Xiang, X.; Grizzle, W.; Sun, D.; Zhang, S.; Axtell, R.C.; Ju, S.; Mu, J.; Zhang, L.; Steinman, L.; et al. Treatment of Brain Inflammatory Diseases by Delivering Exosome Encapsulated Anti-inflammatory Drugs From the Nasal Region to the Brain. Mol. Ther. 2011, 19, 1769–1779, doi:10.1038/MT.2011.164.
  2. Devraj, K.; Guérit, S.; Macas, J.; Reiss, Y. An In Vivo Blood-brain Barrier Permeability Assay in Mice Using Fluorescently Labeled Tracers. J. Vis. Exp 2018, 57038, doi:10.3791/57038.

Introduction:

  1. A/some reference/s to explain the possible entry of EVs through the BBB would be appreciable. 

We have mentioned the possible entrance patways in Introduction lines 72-74, and covered in more detail in Discussion lines 519-532.

  1. Some recent studies or reviews can be cited to define exosomes, especially since 'exosome' is a subgroup of the small EVs fraction (Saint-Pol et al., 2020 Cells; Kowal et al., 2016 PNAS; Mathieu et al., 2019 Nat Cell Biol; 2021 Nat Comm; Théry et al., 2018 JEV). Please be careful also all along the manuscript regarding that point.

We have included the endosomal origin as an additional exosome descriptor in line 76. We also took into account the advice to use the term more carefully.

Results:

  1. Figure 2 and A1 can be melted. 

We have merged the images characterising the EVs from rat and mice airway cultures together (Figure 2 and Figure A3), as, we guess, this was expected?

  1. Concerning the 'exosome' characterization, morphology and size fit with small EVs, but only 2 tetraspanins as 'positive markers' is clearly not enough. Please include other 'positive markers' as proves of their endosomal origin (syntenin-1, Tsg101) and some 'negative markers' to complete your demonstration, such as calnexin, HSP70. Without these complementary experiments, the name 'exosome' is excluded. Please see Thery et al., 2018 (JEV) to follow the updated Minimal Instructions for the study of Extracellular Vesicles (MISEV).

We agree with the remark, and we have changed the definition of our preparation to exosome-like vesicles (ELVs).

  1. Conclusions need to be measured, and not affirmed since the demonstration is slight for some points.

The conclusions are revised to a more moderate version.

  1. Reference 7 is not well formatted in the list.

The format was corrected; after adding some citations, it is reference 10.  

Round 2

Reviewer 2 Report

The authors reviewed the manuscript with care and switched the term 'exosome' into 'exosome-like vesicle' to better fit with the EV terminology. Despite some blots which would be nice to be done to fix the endogenous origin of ELVs, this paper is acceptable in the present form.